# The Prevalence of Virulence Factor Genes among Carbapenem-Non-Susceptible *Acinetobacter baumannii* Clinical Strains and Their Usefulness as Potential Molecular Biomarkers of Infection

**DOI:** 10.3390/diagnostics13061036

**Published:** 2023-03-08

**Authors:** Dagmara Depka, Tomasz Bogiel, Mateusz Rzepka, Eugenia Gospodarek-Komkowska

**Affiliations:** 1Microbiology Department, Ludwik Rydygier Collegium Medicum in Bydgoszcz, Nicolaus Copernicus University in Toruń, 85-094 Bydgoszcz, Poland; 2Department of Clinical Microbiology, Antoni Jurasz University Hospital No. 1, 85-094 Bydgoszcz, Poland

**Keywords:** *Acinetobacter baumannii*, *bla*_OXA_ genes, carbapenem-resistance, HAI, virulence factor genes

## Abstract

Healthcare-associated infections caused by multidrug-resistant *Acinetobacter baumannii* strains are a serious global threat. Therefore, it is important to expand the knowledge on the mechanisms of pathogenicity of these particular bacteria. The aim of this study was to assess the distribution of selected virulence factor genes (*bap*, *surA1*, *omp33-36*, *bauA*, *bauS*, and *pld*) among carbapenem-non-susceptible clinical *A. baumannii* isolates and to evaluate their potential usefulness as genetic markers for rapid diagnostics of *A. baumannii* infections. Moreover, we aimed to compare the virulence genes prevalence with the occurrence of carbapenemases genes. A total of 100 carbapenem-non-susceptible *A. baumannii* clinical isolates were included in the study. The presence of virulence factors and *bla*_OXA_ genes was evaluated by real-time PCR. The occurrence of virulence factors genes was as follows: 100.0% for the *bap* and *surA1* genes, 99.0% for the *basD* and *pld* genes. The *bauA* and *omp33-36* genes were absent among the studied strains. The predominant genes (*bap* and *surA1*) are involved in biofilm formation and their presence among all clinical strains can be applied as a genetic marker to recognize *A. baumannii* infection. High frequencies of the *basD* gene—involved in siderophore biosynthesis and the gene encoding phospholipase D (*pld*)—were also noted among *bla*_OXA_-positive strains, showing their potential role in a pathogenicity of *bla*_OXA_-positive *A. baumannii* clinical strains.

## 1. Introduction

*Acinetobacter baumannii* are non-fermenting, aerobic Gram-negative rods [1]. Due to their ability to survive in a hospital environment, these bacteria mainly cause healthcare-associated infections (HAIs) [2]. Ventilator-associated pneumonia (VAP) and bacteriaemia are the most common infections caused by them, with highest mortality rates [3].

An increasing threat of *A. baumannii* infections is related to a high percentage of antimicrobial-resistant isolates. Therefore, Rice et al. [4] included *A. baumannii* representatives into a group of bacteria known as ESKAPE (among *Enterococcus faecium*, *Staphylococcus aureus*, *Klebsiella pneumoniae*, *Pseudomonas aeruginosa*, and *Enterobacter* spp.). This acronym describes bacteria that can easily avoid the effects of antimicrobials treatment. Moreover, in 2017, the World Health Organization recognized carbapenem-resistant *A. baumannii* (CRAb) as critical pathogens for which profound research and development of effective drugs are needed [5].

*A. baumannii* strains possess many intrinsic resistance mechanisms. Moreover, due to the plasticity of their genome, they also easily acquire new mechanisms, such as genes encoding antibiotic hydrolyzing enzymes [6], with carbapenem-hydrolyzing class D beta-lactamases (CHDLs, oxacillinases) prevailing among these bacteria. Among the aforementioned enzymes, the most prevalent oxacillinases families are chromosomally encoded OXA-51-like and the following enzymes encoded on mobile genetics elements (plasmids, transposons): OXA-23-like, OXA-40/24-like, OXA-58-like, OXA-143-like, and OXA-235-like carbapenemases [7].

The pathogenicity of *A. baumannii* is determined by factors involved in the following processes: bacteria transmission, binding to host structures, human cells damage and invasion [8]. In turn, bacterial viability in the hospital environment is associated the most with *A. baumannii* biofilm formation. Proteins and structures involved in biofilm formation are mainly: pili, outer membrane proteins (surface antigen protein 1—SurA1), and adhesins (biofilm-associated protein—Bap). As it has been previously shown, Bap is involved in intercellular adhesion, ensuring the maturation of biofilm on biotic and abiotic surfaces, with mutations in the *bap* gene causing a weakening of adhesive capacity of *A. baumannii* strains [9].

Bacterial porins are involved in adhesion and they have also been shown to play a crucial role in an apoptosis process. The 33-36-kDa Omp protein (Omp 33-36) is a water channel in *A. baumannii* cells and plays a role in apoptosis and modulates autophagy. The Omp 33-36 is released into the host cells inside the outer membrane vesicles. This protein has also been shown to have an effect on DNA fragmentation in HeLa and Hep-2 cells lines [10,11].

Bacterial cells, during the proceeding infection, have limited access to iron, which is important for their viability. Most bacteria, including *A. baumannii*, obtain iron within siderophores. Acinetobactin is the most important siderophore among *A. baumannii* strains [1]. BauS and BauA proteins are needed for acinetobactin biosynthesis and transport, respectively. These proteins have also been shown to play a vital role in the apoptotic death of epithelial cells [12].

Phosphatidylcholine (PC) is one of the phospholipids that builds the cell membrane of eukaryotic cells; among lung structures, almost 80% of all phospholipids are PC compounds, which is very important due to the localization of the majority of infections (VAP). *A. baumannii* cells synthesize phospholipases (e.g., phospholipase D—PLD) that hydrolyze PC and alter the permeability of the host cell membrane [13,14].

The investigated virulence genes are all chromosomally encoded and play a role inbacterial viability. In the relevant literature, there is a lack of information on the transfer of these genes between species, except for the plasmid-encoded *bla_OXA_* genes, which can potentially be transferred between bacteria [7].

The aims of the study were to: assess the frequency of the selected virulence genes presence (*bap*, *surA1*, *omp33-36*, *bauA*, *bauS*, and *pld*) involved in pathogenicity of CRAb clinical strains and to evaluate their potential usefulness as genetic markers for rapid diagnostics of *A. baumannii* infections. The objective of the study was also to investigate the prevalence of *bla*_OXA_ genes among these strains and the occurrence of virulence factor genes among strains carrying different oxacillinases genes.

## 2. Materials and Methods

### 2.1. Bacterial Isolates and Their Origin

The study included 100 CRAb clinical strains. They were all isolated in the Clinical Microbiology Department from different patients hospitalized at University Hospital No. 1 in Bydgoszcz, Poland, between 2017 and 2020. In total, 63 (63.0%) strains were derived from the patients of the Anesthesiology and Intensive Care Unit (for the detailed origin of the strains, see Appendix A).

*A. baumannii* strains were isolated from various clinical specimen types, mainly from respiratory tract infections (RTIs) (*n* = 46) (for the detailed data, see Appendix A).

The studied strains were identified using MALDI-TOF MS (Bruker, Mannheim, Germany) in a routine laboratory diagnostic procedure according to the manufacturer’s protocol. The MALDI Biotyper software version 4.2.28 was used for bacterial identification. The cut-off scores for identification were as follows: ≥1.7—a reliable identification to the genus level, rate ≥ 1.7—a reliable identification to the species level. In the current study, scores above 2.0 were reached for every strain. The bacterial test standard (BTS, Bruker, Mannheim, Germany) was used as identification quality control.

The isolates were stored in Brain Heart Infusion (Becton, Dickinson and Company, Sparks, MD, USA) with 15% glycerol (Sigma-Aldrich, St. Louis, MO, USA) at −80 °C and were grown on MacConkey Agar (Becton, Dickinson and Company, Sparks, MD, USA) before the testing.

### 2.2. Antimicrobial Susceptibility Testing

Antimicrobial susceptibility testing (AST) of the studied isolates was performed with BD Phoenix™ M50 NMIC-402 panels (Becton, Dickinson and Company, Sparks, MD, USA) in a routine laboratory diagnostic scheme. The results of AST were interpreted according to the European Committee on Antimicrobial Susceptibility Testing recommendation v 13.0 (EUCAST). *P. aeruginosa* ATCC 27853 and *Escherichia coli* ATCC 25922 were used as quality control strains, according to EUCAST recommendations [15].

The classifications of the strains into multidrug-resistant (MDR) and pandrug-resistant (PDR) groups were as follows: MDR—strains resistant to at least three of the tested antimicrobials belonging to separate antibiotic groups, and PDR—strains resistant to all of the tested antimicrobials [16].

### 2.3. DNA Extraction, Virulence Factor, and bla_OXA_ Genes Detection

DNA samples from the investigated *A. baumannii* strains were isolated using the Genomic Mini kit (A&A Biotechnology, Gdansk, Poland) according to the manufacturer’s protocol. DNA was stored at −20 °C before the examination.

The virulence factors and *bla*_OXA_ genes were detected by real-time PCR with the LightCycler 480 II (Roche, Basel, Switzerland) and CFX OPUS 96 (Bio-Rad, Hercules, CA, USA). The specification of primers used in the study is summarized in Table 1. Final volume of each reaction was 20 µL. The reagents were used as follows: 4 µL of 5 × HOT FIREPOL^®^ EvaGreen^®^ Mix (SolisBiodyne, Tartu, Estonia), 5 µL of molecular biology grade water (EurX, Gdansk, Poland), 5 µL (0.25 µM/reaction) of each primer (Genomed, Warsaw, Poland), and 1 µL of the DNA extracted from the investigated isolates. The following temperature conditions were applied: one cycle at 95 °C for 12 min, followed by 40 cycles at 95 °C for 15 s, annealing temperature (depending on the detected gene, see Table 1) for 20 s, and the final elongation—72 °C for 20 s. The data collection was enabled at each extension step. The melt curve protocol was added afterwards—95 °C for 5 s and 60 °C for 1 min—followed by data acquisition at 0.11 °C increments between 60 °C and 97 °C to confirm the specificity of the amplification product (for the results, see Appendix A). The data collection was enabled continuously at each increment during the temperature change (high-resolution melting).

The DNA extracted form *A. baumannii* DSMZ (Deutsche Sammlung von Mikroorganismen und Zellkulturen, Germany) 30008 and *A. baumannii* DSMZ 102930 were used as the reference strains (positive controls) for the investigated genes presence (the distribution of virulence factors genes among the references *A. baumannii* strains are shown in Appendix A). The molecular biology grade water (EurX, Gdansk, Poland) was used as a negative control (NTC).

### 2.4. Statistical Data Analysis

Statistical analysis was performed using the Statistica™ 13.3 (TIBCO Software Inc., Palo Alto, CA, USA) program, using Spearman’s rank-order correlation test to investigate the correlation for particular genes’ coexistence and the strains origin (*p* ≤ 0.05). An interpretation of Spearman’s rank-order correlation test was as follows: for *r* < 0.2—no correlation observed; 0.2–0.4—weak correlation; 0.4–0.7—moderate correlation; 0.7–0.9—strong correlation; >0.9—very strong correlation.

## 3. Results

### 3.1. Antimicrobial Susceptibility

All the tested *A. baumannii* isolates were non-susceptible to carbapenems and showed a high percentage of resistance to most of the antimicrobial agents tested. Except for colistin, the in vitro activity of other antimicrobial agents was relatively low (with about 20% strains susceptible to aminoglycosides). All of the strains belonged to MDR, while 6 (6.0%) were additionally PDR. Antimicrobial susceptibility results of the studied strains are shown in Table 2.

### 3.2. bla_OXA_ Genes Presence

Among the studied strains, *bla*_OXA-40_ and *bla*_OXA-23_ were detected in 62 (62.0%) and 35 (35.0%) isolates, respectively. Very strong negative correlation was revealed between the occurrence of both genes (*r* = −0.937306).

*bla*_OXA_ genes were not detected in three (3.0%) isolates. These were the only CRAb strains without already known carbapenemases that were available during the study and therefore included in the study. One of them was susceptible, with increased exposure to imipenem and meropenem, while the second one was susceptible, with increased exposure to imipenem and susceptible to meropenem. The distribution of susceptibility profiles for the investigated strains is shown in Table 3. Twenty antimicrobial susceptibility profiles were distinguished. The profile of susceptibility to colistin was the only one (44.0%) that was most prevalent, regardless of the *bla*_OXA_ genes detected.

There was a weak positive correlation between the presence of the *bla*_OXA-23_ gene and the strains’ origin from RTI cases (*r* = 0.24819). The *bla*_OXA-40_-positive isolates were derived statistically more often from other sites of infections—very weak positive correlation (*r* = 0.22818). The detailed prevalence of the *bla*_OXA_ genes in *A. baumannii* in relation to the strains’ origin is shown in Table 4.

### 3.3. Virulence Factor Genes Presence and Co-Existence

The prevalence of virulence factors genes was as follows: 100 (100.0%) for the *bap* and *surA1* genes, 99 (99.0%) for the *basD* and *pld* genes. The *bauA* and *omp33-36* genes were not detected among the studied strains. Table 5 shows the occurrence of virulence factors genes with relation to the detected carbapenemase gene.

There were no statistically significant differences between the detected carbapenemase gene and the particular virulence factor gene presence (Spearman’s rank-order correlation below 0.2). The only difference was shown for the two *bla*_OXA-23_-positive isolates, one of which was absent for the *basD* gene, while the other lacked the *pld* gene.

## 4. Discussion

HAIs caused by CRAb are a global problem due to limited therapeutic options for their treatment. *A. baumannii* strains are persistent in the hospital environment due to the development of resistance mechanisms to disinfection and a high survival rate in dry environments [22]. The CRAb strains account for almost 90% of *A. baumannii* infections in some countries, mostly in southern Europe and Asia [23]. Moreover, the European Center for Disease Prevention and Control report in 2021 highlights the increased isolation rate of *Acinetobacter* spp., mainly *A. baumannii* complex representatives, during the two years of the pandemic (2020–2021). The reason might have been that more patients were mechanically ventilated in the aforementioned time period due to Coronavirus disease [24].

*A. baumannii* strains frequently exhibit antimicrobial susceptibility profiles categorized as MDR or even PDR. In our study, six (6.0%) of the isolates were PDR. Resistance to carbapenems is often connected with lack of susceptibility to other groups of antibiotics (e.g., fluoroquinolones) [25]. In our study, 100% of CRAbs were simultaneously resistant to fluoroquinolones, only about 20% of isolates were susceptible to aminoglycosides, which correlates with the reports of other researchers [26,27]. Therefore, the only treatment option for the infections with this etiology is often colistin [28]. This is in concordance with the results of our study, with the most prevalent antimicrobial susceptibility profile consisting of the strains exclusively susceptible to colistin.

The overuse of antibiotics, such as carbapenems, results in CRAb selection [29]. The most prevalent mechanisms of carbapenem resistance among *A. baumannii* isolates are CHDLs, with OXA-23 occurring the most often among *A. baumannii* worldwide [30,31]. Meanwhile, the results of our study showed that in our region, OXA-40-positive isolates are dominant, which is consistent with other studies conducted in Poland [32]. In our study, no isolates were simultaneously positive for both tested *bla*_OXA_ genes.

The main factor in the early stages of pathogenicity and persistence of *A. baumannii* strains is biofilm formation. Bap is a factor involved in biofilm synthesis, but also in an adhesion to epithelial cells and abiotic surfaces [33]. The results of our study show that all *A. baumannii* isolates carried the *bap* gene, which makes this gene useful as one of the markers of *A. baumannii* infections. Our findings are similar to the results obtained by Monfared et al. [34]. They have also shown that the bacteria lacking the *bap* gene are incapable of biofilm synthesis. Of note, the isolates used in this study were derived from specific clinical specimens (mostly broncho-alveolar lavage and wound swabs) where the biofilm formation is crucial for an infection initiation.

The outer membrane protein also involved in biofilm formation, adhesion, and serum resistance is a surface antigen protein 1 (*surA1*) [18]. Liu et al. described that 100.0% of MDR and 83.3% of non-MDR *A. baumannii* isolates carry the *surA1* gene [20]. The results of the aforementioned study are similar with our observations; all the tested isolates were positive for the *surA1* gene, also making this gene a potential biomarker of infections with this etiology.

Interestingly, due to the fact that the presence of the *bap* and *surA1* genes is noted in all of the clinical strains, they can be altogether used as a genetic marker to confirm *A. baumannii* infection, with an application using molecular biology methods, however, this observation requires further research.

Outer membrane protein 33-36 (*omp 33-36*) is involved in adherence, invasion, and cytotoxicity to human epithelial cells [10,11]. The previous study showed that the decrease expression of Omp33-36 could be also associated with imipenem resistance [35]. None of the *A. baumannii* tested in this study possessed this gene, but all were non susceptible to imipenem. In turn, other researchers have shown that there is no statistically significant difference in the frequency of this gene among CARb and non-CRAb strains [20].

Phospholipase D is an enzyme involved in serum resistance and invasion due to hydrolysis of phospholipids which build the host cell membrane [13]. Phospholipase D cleaves off a head group of phospholipids and could interfere with the immune system response [1]. The results of our study indicate a high prevalence of the *pld* gene (99.0%) among CRAb strains, which is similar to the results of the previous study [20]. Therefore, it has been concluded that phospolipase D may play an important role in the pathogenesis of *A. baumannii* infections.

The system of iron acquisition in *A. baumannii* is based on the production of low-molecular iron-specific chelators—siderophores. The best known siderophore, produced by *A. baumannii* strains, is acinetobactin. The biosynthesis of acinetobactin is accomplished by the proteins BasA–D and BasF–J [36]. Siderophores enter the bacterial cell in an active transport way, for which appropriate receptors on the cell membrane are needed, e.g., BauA is a receptor for acinetobactin. One of the previous studies showed a high prevalence of gene encoding BauA (*bauA*) especially among MDR isolates [37]. Our results are contradictory—no isolates with the *bauA* gene were detected. In turn, the *basD* gene was detected in 99.0% of isolates and this is in line with the results of other studies, showing *basD* more often (92.0%) than *bauA* (62.5%) among MDR isolates [20]. Porbaran et al. [38] showed the prevalence of *basD* and *bauA* in a lower percentage of *A. baumannii* isolates among 12.5% and 15.2%, respectively. This is more concordant with the results of our study. They also showed a strong correlation between the prevalence of metallo-beta-lactamases and AmpC genes and the occurrence of genes involved in the iron/siderophore uptake system. The genes encoding the acinetobactin metabolism protein are normally located within the Acinetobactin Cluster, meanwhile Eijkelkamp et al. [39] described that the *A. baumannii* SDF isolate lacks the Acinetobactin gene cluster. This may be one of the explanations why the *bauA* gene was not detected among any of the isolates included into our study.

All the investigated genes encode proteins involved in the viability and pathogenicity of *A. baumannii*. In this study, we only examined the pathogenic potential of the tested strains, however, evaluating the presence of these genes among clinical *A. baumannii* isolates gives better insight into their molecular characteristics. A potential importance of particular genes as biomarkers of these pathogens was also underlined, however, more research is needed to investigate the actual significance of these virulence factors, for example, the ability to biofilm synthesis or assessing the expression levels of the genes studied.

A limitation of this study was that only carbapenem-non-susceptible *A. baumannii* strains were considered. However, they are definitely the most prevalent isolates worldwide and therefore the most promising study objects. Nevertheless, in order to complete the data on the distribution of virulence factor genes, it would be preferable to investigate carbapenem-susceptible isolates also. However, due to such widespread presence of MDR and PDR *A. baumannii* isolates, the multidrug-susceptible strains are currently not easily reachable.

To the best of our knowledge, this is the first work showing the prevalence of virulence factors genes in terms of the presence of *bla*_OXA_ genes among *A. baumannii* isolates derived from patients hospitalized in Poland. Definite further studies on this issue are necessary, e.g., genes presence confirmation by DNA sequencing, testing for more clinically threatening beta-lactamases such as OXA-48 or NDM-1. However, there are no statistically significant differences or obvious correlations between the presence of virulence factor genes and *bla*_OXA_ genes. Comparing the result of our study to the studies of other researchers, it can be concluded that the *omp33-36* and *bauA* genes are not essential for the pathogenesis of infections caused by *bla*_OXA_-positive *A. baumannii* clinical isolates, at least for the infections localized in the sites considered in the present study.

## 5. Conclusions

The most prevalent virulence factors genes among studied *A. baumannii* isolates are the *bap* and *surA1* genes. These genes, especially their co-existence, could be used as biomarkers for the diagnostics of *A. baumannii* infections. A high percentage of the gene encoding a protein involved in the biosynthesis of the siderophore (acinetobactin)—*basD* is also observed among *A. baumannii*. Meanwhile, the *bauA* gene—encoding the transport protein for acinetobactin is absent among carbapenem-non-susceptible isolates. Therefore, there is a possibility of another pathway for the siderophore transport for this particular group of *A. baumannii* strains.

## Figures and Tables

**Table 1 diagnostics-13-01036-t001:** Sequences of primers applied for virulence genes detection and real-time PCR parameters used in the study.

Gene Detected	Primer Sequences 5′→3′	Tm (°C)	Annealing Temperature (°C)	References
*bap*	F: AGTTAAAGAAGGGCAAGAAG	47.7	58	[17]
	R: GGAGCACCACCTAACTGA	50.3		
*surA1*	F: CAATTGGTAGCTGGCGATCA	51.8	58	[18]
	R: TTAGGCGGGACTCAGCTTTT	51.8		
*basD*	F: CTCTTGCATGGCAACACCAC	53.8	60	[19]
	R: CCAACGAGACCGCTTATGGT	53.8		
*bauA*	F: TGGCAAGGTGAAAATGCACG	51.8	60	[20]
	R: GCCGCATATGCCATCAACTG	53.8		
*pld*	F: CCGTCAATTACGCCAAGCTG	53.8	60	[13]
	R: CTGACGCTACCTGACGGTTT	53.8		
*omp33-36*	F: ATTAGCCATGACCGGTGCTC	53.8	60	[10]
	R: CCACCCCAAACATGGTCGTA	53.8		
*bla* _OXA-40_	F: GGTTAGTTGGCCCCCTTAA	51.8	58	[21]
	R: AGTTGAGCGAAAAGGGGATT	49.7		
*bla* _OXA-23_	F: GATCGGATTGGAGAACCAGA	50.3	58	[21]
	R: ATTTCTGACCGCATTTCCAT	47.7		

F—forward primer, R—reverse primer, Tm—primer melting temperature.

**Table 2 diagnostics-13-01036-t002:** Antimicrobial susceptibility of the studied strains (*n* = 100).

	Number (%) of Strains		
Antimicrobial	Susceptible	Susceptible, Increased Exposure	Resistant
Imipenem (IPM)	0 (0.0)	2 (2.0)	98 (98.0)
Meropenem (MEM)	2 (2.0)	1 (1.0)	97 (97.0)
Gentamicin (GEN) ^1^	21 (21.9)	0 (0.0)	75 (78.1)
Amikacin (AMK)	17 (17.0)	5 (5.0)	78 (78.0)
Tobramycin (NN)	22 (22.0)	0 (0.0)	78 (78.0)
Ciprofloxacin (CIP)	0 (0.0)	0 (0.0)	100 (100.0)
Levofloxacin (LEV)	0 (0.0)	0 (0.0)	100 (100.0)
Trimethoprim/sulfamethoxazole (SXT)	2 (2.0)	0 (0.0)	98 (98.0)
Colistin (COL)	90 (90.0)	0 (0.0)	10 (10.0)

^1^ susceptibility to gentamicin was tested on 96 isolates.

**Table 3 diagnostics-13-01036-t003:** Distribution of susceptibility profiles and oxacillinases genes among *A. baumannii* strains included into the study (*n* = 100).

Profile	IPM	MEM	GEN ^1^	AMK	NN	CIP	LEV	SXT	COL	*bla* _OXA_	Number (%) of Strains (*n* = 100)
I	R	R	R	R	R	R	R	R	S	*bla* _OXA-23_	24 (24.0)
II	R	R	R	R	R	R	R	R	S	*bla* _OXA-40_	20 (20.0)
III	R	R	S	R	R	R	R	R	S	*bla* _OXA-40_	10 (10.0)
IV	R	R	R	S	S	R	R	R	S	*bla* _OXA-40_	9 (9.0)
V	R	R	S	S	S	R	R	R	S	*bla* _OXA-40_	4 (4.0)
VI	R	R	R	I	R	R	R	R	S	*bla* _OXA-40_	4 (4.0)
VII	R	R	R	R	S	R	R	R	S	*bla* _OXA-23_	4 (4.0)
VIII	R	R	S	R	R	R	R	R	S	*bla* _OXA-23_	4 (4.0)
IX	R	R	R	R	R	R	R	R	R	*bla* _OXA-23_	3 (3.0)
X	R	R	R	R	R	R	R	R	R	*bla* _OXA-40_	3 (3.0)
XI	R	R	R	S	S	R	R	R	R	*bla* _OXA-40_	2 (2.0)
XII	R	R	R	R	R	R	R	R	S	-	1 (1.0)
XIII	R	R	R	R	R	R	R	S	S	*bla* _OXA-40_	1 (1.0)
XIV	R	R	R	S	S	R	R	R	S	*bla* _OXA-23_	1 (1.0)
XV	R	R	R	R	S	R	R	R	S	*bla* _OXA-40_	1 (1.0)
XVI	R	R	S	R	R	R	R	R	R	*bla* _OXA-40_	1 (1.0)
XVII	I	I	R	R	R	R	R	R	S	-	1 (1.0)
XVIII	I	S	R	R	R	R	R	R	S	-	1 (1.0)
XIX	R	S	S	S	S	R	R	R	R	*bla* _OXA-40_	1 (1.0)
XX	R	R	S	I	R	R	R	R	S	*bla* _OXA-40_	1 (1.0)
XXI ^1^	R	R		R	R	R	R	R	S	*bla* _OXA-23_	3 (3.0)
XXII ^1^	R	R		R	R	R	R	S	S	*bla* _OXA-40_	1 (1.0)

^1^ susceptibility to gentamicin was tested on 96 isolates, R—resistant (pink boxes), I—susceptible, increased exposure (yellow boxes), S—susceptible (green boxes), IPM—imipenem, MEM—meropenem, GEN—gentamicin, AMK—amikacin, NN—tobramycin, CIP—ciprofloxacin, LEV—levofloxacin, SXT—trimethoprim/sulfamethoxazole, COL—colistin.

**Table 4 diagnostics-13-01036-t004:** The occurrence of *bla*_OXA_ genes in *A. baumannii* with respect to strains’ origin.

Clinical Material	*bla* _OXA-40_	*bla* _OXA-23_	*bla*_OXA-40/23_-Negative	Total (*n* = 100)
RTI	23 (50.0%)	22 (47.8%)	1 (2.2%)	46 (46.0%)
Other origin ^1^	39 (72.2%)	13 (24.1%)	2 (3.7%)	54 (54.0%)

^1^ wound infections/pus, invasive infections, urinary tract infections, body fluids, tissue, granulation tissue, vascular catheter (for detailed data, see Appendix A).

**Table 5 diagnostics-13-01036-t005:** The occurrence of virulence factor genes among *A. baumannii* strains included in the study (*n* = 100).

	Number (%) of Strains with a Particular Virulence Factor Gene					
	*bap*	*surA1*	*basD*	*bauA*	*pld*	*omp33-36*
*bla*_OXA-40_ (*n* = 62)	62 (100.0)	62 (100.0)	62 (100.0)	0 (0.0)	62 (100.0)	0 (0.0)
*bla*_OXA-23_ (*n* = 35)	35 (100.0)	35 (100.0)	34 (97.1)	0 (0.0)	34 (97.1)	0 (0.0)
*bla*_OXA-40/23_-negative (*n* = 3)	3 (100.0)	3 (100.0)	3 (100.0)	0 (0.0)	3 (100.0)	0 (0.0)

## Data Availability

The data presented in this study are available on a reasonable request from the corresponding author.

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
