# Peer review of "The Prevalence of Virulence Factor Genes among Carbapenem-Non-Susceptible Acinetobacter baumannii Clinical Strains and Their Usefulness as Potential Molecular Biomarkers of Infection"

_diagnostics, 2023, doi:10.3390/diagnostics13061036_

Round 1

Reviewer 1 Report

I have found no problems with the research reported here. 

I suggest to modulate the article more to the scope of the journal, which is diagnostics. The use of described common genes as biomarkers seem to be added to the manuscript at a later timepoint, possibly to re-submit to this journal. However,  ideally it should already be highlighted in the abstract that the purpose of the study was at least partially to find diagnostic markers. 

minor:

please thoroughly correct grammar

some paragraphs are printed in grey instead of black

Author Response

We would like to thank the honorable Reviewer for carefully reading of the manuscript and providing all the valuable comments. Please see below a detailed point-by-point response to all your comments.

  1. I suggest to modulate the article more to the scope of the journal, which is diagnostics. The use of described common genes as biomarkers seem to be added to the manuscript at a later timepoint, possibly to re-submit to this journal. However, ideally it should already be highlighted in the abstract that the purpose of the study was at least partially to find diagnostic markers. - It has been underlined.
  2. please thoroughly correct grammar - It has been corrected.
  3. some paragraphs are printed in grey instead of black - It has been corrected.

Reviewer 2 Report

1. Carbapenem-insensitive strains in the manuscript can be changed to carbapenem-resistant strains.

2. Line 104: AST results need to be controlled with quality control strains.

3. Whether these virulence factor genes are mobile, and whether they can be transferred to other strains to produce virulence.

4. Do these virulence genes express virulence?

Author Response

We would like to thank the honorable Reviewer for carefully reading of the manuscript and providing all the valuable comments. Please see below a detailed point-by-point response to all your comments.

  1. Carbapenem-insensitive strains in the manuscript can be changed to carbapenem-resistant strains. – We want to thank the Reviewer for this remark especially. However, two of all the isolates tested (with imipenem MICs of 4 mg/l) were classified, according to EUCAST recommendation, as susceptible with increased exposure. Therefore, not all the isolates were in fact resistant to carbapenems.
  2. Line 104: AST results need to be controlled with quality control strains. – The information on quality control strains used in the study has been added.
  3. Whether these virulence factor genes are mobile, and whether they can be transferred to other strains to produce virulence. – The following sentences have been added into the manuscript: The investigated virulence genes are chromosomally encoded and play a role in a bacterial viability. In the relevant literature there is a lack of information on the transfer of these genes between species, except for the plasmid-encoded blaOXA genes, which can potentially be transferred between bacteria
  4. Do these virulence genes express virulence? – We want to thank the Reviewer for this remark especially. We are truly interested in the further research focused on this aspect. The following sentences have been added into the manuscript: All the investigated genes encode proteins involved in the viability and pathogenicity of A. baumannii. In this study, we only examined the pathogenic potential of the tested strains, evaluating the presence of these genes among clinical A. baumannii isolates as their molecular characteristics. A potential importance of particular genes as biomarkers of these pathogens was also discussed but more research is needed to investigate the actual significance of these virulence factors, for example, ability to biofilm synthesis or assessing the expression levels of the genes studied.

Reviewer 3 Report

The problem of infections caused by Acinetobacter baumanii is serious and involves the efforts of many centres around the world. The WHO has declared this pathogen a priority against which rapid therapeutic solutions are needed, especially in view of the increasing drug resistance of strains found in healthcare settings. The aim of this study was to assess the presence of six virulence genes and three beta-lactamase synthesis determinants, OXA-23 and OXA-40, which confer beta-lactam resistance. To their credit, up to 100 clinical strains of A. baumanii were isolated between 2017 and 2020, which is very worrying. The authors used modern testing techniques, including MALDI-TOF MS mass spectrometry and the Phoenix system used in routine diagnostics, to identify and assess the presence of selected virulence genes.

However, several shortcomings in the drafting of the manuscript were not shied away from by the authors, including:
1. the drug susceptibility of the strains included in the study should have been assessed according to the latest recommendations, i.e. EUCAST Clinical Breakpoint Tables v. 13.0, valid from 2023-01-01;
2. strain identification was performed by MALDI-TOF MS mass spectrometry, although details of the procedure, parameters and rules of interpretation were not provided;
3. the reference methods for susceptibility testing are the EUCAST standardised disc diffusion method or the broth microdilution method - have the results obtained with the BD Phoenix system been tested against either of these methods?;
4. EUCAST susceptibility testing requires quality control in the form of a Pseudomonas aeruginosa ATCC 27853 reference strain - is this criterion met? Do we have confidence in the quality of these tests?;
5. the genetic tests used a real-time PCR method, but what were the positive and negative controls? this information is missing;
6. a publication by Liu et al, 2018 [17] is cited for the primers used, but this is not the source of the sequences; the authors of this publication did not design these primers;

7. in the description of the results of the drug susceptibility profile, the authors used different names for the criteria - sensitivity and susceptibility are written once; according to the EUCAST definitions, there should be 3 following categories: susceptibility - susceptible, increased exposure - resistant; in the tables and in the text, they are written differently, so please standardise these names;
8. the authors also use different notations for the percentage of strains, i.e. once as in line 152 "....were detected in 62.0% (n=62) and ......". once as in line 154 "in three (3.0%) isolates...", and again differently in the tables; although the headers of Tables 2, 3 and 5 state "number (%)", so it is no longer necessary to state the number with a % sign in them;
I also note the need for some cosmetic corrections:
9. the introduction lacks the addition of information as to which of the genes studied is responsible for the transcription of the proteins described.
10. line 144: "...other antimicrobial agents was realistically low..." this would be better replaced by "relatively low";
11. line 155, in the sentence "They were the only available CRAb strains without already known ....", shouldn't the beginning be "They were..."? it's confusing.
12. in the manuscript, the authors checked for the presence of genes responsible for, among other things, the adhesion process and biofilm formation; it was concluded that the presence of the bap and surA1 genes in 100% of the strains tested indicated the ability to initiate the biofilm formation process - was it phenotypically checked to which groups the tested strains belonged of weak, strong producers? the presence of a gene in the bacterial genome does not always indicate such an ability, which is revealed by the phenotype.
13. In Table 3, instead of listing all the categories of susceptibility of the strains, their profile could be listed after a comma, e.g. I - IPM,MEM,GEN,AMK,NN,CIP,LEV,SXT, then the table would be clearer;
14. Table 4 compares the presence of resistance genes between the diagnostic materials that were the source of the strains tested, perhaps a table could be made with a more precise distinction of the "other materials" described in the Appendix?;

15. In my opinion, the final statement that "this is the first paper showing the prevalence of virulence factor and blaOXA genes among A. baumanii isolates" is exaggerated, except in relation to Poland isolates, because the presence of these genes used in this study, but also of many other beta-lactamase genes in MDR, PDR or XDR strains of these bacteria, has already been studied in many publications all over the world and is not surprising; moreover, in relation to such a dangerous pathogen, their presence should be confirmed by sequencing at least, as is the case with opportunistic bacteria. Testing for more clinically threatening beta-lactamases such as OXA-48 or NDM-1 may also be considered;

16. The authors point out that the statistical analysis is based on statistical significance, which they highlight in the conclusions, and this parameter is nowhere to be found in the results.

In conclusion, the authors' efforts to collect such a large collection of MDR A. baumanii strains and to assess their drug susceptibility and the occurrence of selected virulence genes and beta-lactamase synthesis determinants should be appreciated. I believe that a better refinement of the article will help to improve the quality of such epidemiologically important results.

Author Response

We would like to thank the honorable Reviewer for carefully reading of the manuscript and providing all the valuable comments. Please see below a detailed point-by-point response to all your comments.

  1. the drug susceptibility of the strains included in the study should have been assessed according to the latest recommendations, i.e. EUCAST Clinical Breakpoint Tables v. 13.0, valid from 2023-01-01; - It has been corrected.
  2. strain identification was performed by MALDI-TOF MS mass spectrometry, although details of the procedure, parameters and rules of interpretation were not provided; - It has been corrected.
  3. the reference methods for susceptibility testing are the EUCAST standardised disc diffusion method or the broth microdilution method - have the results obtained with the BD Phoenix system been tested against either of these methods?; - The BD Phoenix system has been fully validated using the reference methods by the manufacturer, moreover a systematic intra-laboratory antimicrobial susceptibility testing quality control is performed constantly using a reference strains.
  4. EUCAST susceptibility testing requires quality control in the form of a Pseudomonas aeruginosaATCC 27853 reference strain - is this criterion met? Do we have confidence in the quality of these tests?; - We want to thank the Reviewer for this remark especially. Yes, it was our oversight in the initial version of the manuscript and the missing information has been added into the current version.
  5. the genetic tests used a real-time PCR method, but what were the positive and negative controls? this information is missing; - It has been corrected.
  6. a publication by Liu et al, 2018 [17] is cited for the primers used, but this is not the source of the sequences; the authors of this publication did not design these primers; - It has been corrected.
  7. in the description of the results of the drug susceptibility profile, the authors used different names for the criteria - sensitivityand susceptibilityare written once; according to the EUCAST definitions, there should be 3 following categories: susceptibility - susceptible, increased exposure - resistant; in the tables and in the text, they are written differently, so please standardise these names; - It has been corrected.
  8. the authors also use different notations for the percentage of strains, i.e. once as in line 152 "....were detected in 62.0% (n=62) and ......". once as in line 154 "in three (3.0%) isolates...", and again differently in the tables; although the headers of Tables 2, 3 and 5 state "number (%)", so it is no longer necessary to state the number with a % sign in them; - It has been corrected.
  9. the introduction lacks the addition of information as to which of the genes studied is responsible for the transcription of the proteins described. - It has been added.
  10. line 144: "...other antimicrobial agents was realistically low..." this would be better replaced by "relatively low"; - It has been corrected.
  11. line 155, in the sentence "They were the only available CRAb strains without already known ....", shouldn't the beginning be "They were..."? it's confusing. - It has been corrected.
    12. in the manuscript, the authors checked for the presence of genes responsible for, among other things, the adhesion process and biofilm formation; it was concluded that the presence of the bapandsurA1 genes in 100% of the strains tested indicated the ability to initiate the biofilm formation process - was it phenotypically checked to which groups the tested strains belonged of weak, strong producers? the presence of a gene in the bacterial genome does not always indicate such an ability, which is revealed by the phenotype. - We want to thank the Reviewer for this remark especially. We are truly interested in the further research focused on this aspect. The following sentences have been added into the manuscript: All the investigated genes encode proteins involved in the viability and pathogenicity of A. baumannii. In this study, we only examined the pathogenic potential of the tested strains, evaluating the presence of these genes among clinical A. baumannii isolates as their molecular characteristics. A potential importance of particular genes as biomarkers of these pathogens was also discussed but more research is needed to investigate the actual significance of these virulence factors, for example, ability to biofilm synthesis or assessing the expression levels of the genes studied.
  12. In Table 3, instead of listing all the categories of susceptibility of the strains, their profile could be listed after a comma, e.g. I - IPM,MEM,GEN,AMK,NN,CIP,LEV,SXT, then the table would be clearer; - Table 3 was intended to show the differences between strains belonging to different antibiotypes (isolates with different antimicrobial susceptibility profiles) and different blaOXA genes profiles, therefore the table is prepared in this way.
  13. Table 4 compares the presence of resistance genes between the diagnostic materials that were the source of the strains tested, perhaps a table could be made with a more precise distinction of the "other materials" described in the Appendix?; - It has been corrected. The detailed data has been added in supplementary material Table S2.
  14. In my opinion, the final statement that "this is the first paper showing the prevalence of virulence factor and blaOXAgenes among A. baumaniiisolates" is exaggerated, except in relation to Poland isolates, because the presence of these genes used in this study, but also of many other beta-lactamase genes in MDR, PDR or XDR strains of these bacteria, has already been studied in many publications all over the world and is not surprising; moreover, in relation to such a dangerous pathogen, their presence should be confirmed by sequencing at least, as is the case with opportunistic bacteria. Testing for more clinically threatening beta-lactamases such as OXA-48 or NDM-1 may also be considered; - It has been corrected.
  15. The authors point out that the statistical analysis is based on statistical significance, which they highlight in the conclusions, and this parameter is nowhere to be found in the results. – The conclusions emphasized that there were no statistically significant differences or obvious correlations between the presence of virulence factor genes and blaOXA genes – all the statistical data/correlations were calculated are included into the results section.

Round 2

Reviewer 2 Report

In addition to supplementing in the text, the required experiments should continue to be refined

Reviewer 3 Report

I appreciate the authors' efforts and corrections. As it is, the manuscript is much improved and of a higher quality than the original.